# MicroRNA Omics Analysis of *Camellia sinesis* Pollen Tubes in Response to Low-Temperature and Nitric Oxide

**DOI:** 10.3390/biom11070930

**Published:** 2021-06-23

**Authors:** Xiaohan Xu, Weidong Wang, Yi Sun, Anqi Xing, Zichen Wu, Zhiqiang Tian, Xuyan Li, Yuhua Wang

**Affiliations:** 1College of Horticulture, Nanjing Agricultural University, Nanjing 210095, China; 2019204041@njau.edu.cn (X.X.); 2019104085@stu.njau.edu.cn (Y.S.); 2020204039@stu.njau.edu.cn (A.X.); 2018104086@njau.edu.cn (Z.W.); 2019804212@stu.njau.edu.cn (Z.T.); 2019804211@njau.edu.cn (X.L.); 2College of Horticulture, Northwest A&F University, Yangling 712100, China; wangweidong@nwafu.edu.cn

**Keywords:** *Camellia sinensis*, pollen tube growth, microRNA, nitric oxide, low-temperature

## Abstract

Nitric oxide (NO) as a momentous signal molecule participates in plant reproductive development and responds to various abiotic stresses. Here, the inhibitory effects of the NO-dominated signal network on the pollen tube growth of *Camellia sinensis* under low temperature (LT) were studied by microRNA (miRNA) omics analysis. The results showed that 77 and 71 differentially expressed miRNAs (DEMs) were induced by LT and NO treatment, respectively. Gene ontology (GO) analysis showed that DEM target genes related to microtubules and actin were enriched uniquely under LT treatment, while DEM target genes related to redox process were enriched uniquely under NO treatment. In addition, the target genes of miRNA co-regulated by LT and NO are only located on the cell membrane and cell wall, and most of them are enriched in metal ion binding and/or transport and cell wall organization. Furthermore, DEM and its target genes related to metal ion binding/transport, redox process, actin, cell wall organization and carbohydrate metabolism were identified and quantified by functional analysis and qRT-PCR. In conclusion, miRNA omics analysis provides a complex signal network regulated by NO-mediated miRNA, which changes cell structure and component distribution by adjusting Ca^2+^ gradient, thus affecting the polar growth of the *C. sinensis* pollen tube tip under LT.

## 1. Introduction

Low temperature (LT) is one of the momentous factors restricting plant growth and development, geographical distribution and yield expansion, and is reflected in germination, growth, development, flowering and colonization [1]. Particularly, LT often negatively regulates the reproduction process which is an indispensable process in the growth of plants, especially pollen tube germination and elongation [2]. Some earlier studies have indicated that LT reduced the germination and elongation of pollen tubes in various plants such as tobacco [3], hazelnut [2] and pear [4]. *Camellia sinensis* (L.) O. Kuntze as a worldwide beverage crop is also significantly restricted in growth, yield, quality and distribution by LT. A large number of in-depth investigations about the effects of LT on the vegetative organizations of *C. sinensis* have been published recently and they involve cell morphology, physiological and metabolic responses and gene regulation [5,6,7,8,9,10]. Since the flowering and reproduction process of *C. sinensis* are at the end of autumn, they often experience a sharp drop in temperature. Thus, the influence of LT on *C. sinensis* pollen tubes cannot be ignored. Recently, Wang et al. [11] and Wang et al. [12] both reported that LT resulted in a decrease in *C. sinensis* pollen germination rate and pollen tube length, as well as deformed *C. sinensis* pollen tubes. Another study from Çetinbaş-Genç et al. also pointed out that LT caused the abnormal growth of *C. sinensis* pollen tubes [11]. In addition, they all reported a sharp increase in ROS levels and a sharp drop in pH in *C. sinensis* pollen tubes under LT conditions. These researches indicate that the LT resistance mechanism of the *C. sinensis* pollen tube is worthy of further study.

Nitric oxide (NO), an essential signal molecule involved in physiological and pathological processes, plays prominent roles in plants, including but not limited to mitochondrial activity, seed germination, leaf development and fruit maturation [12,13]. Additionally, earlier studies also reported that NO is related to plant reproductive processes such as flower bud differentiation, flowering induction and, especially, tip growth of the pollen tube [14,15]. In recent years, dose-dependent inhibition of NO donors on germination and elongation of the pollen in angiosperms has been reported [16,17]. For example, the NO signaling pathway can regulate extracellular nucleotides to inhibit pollen germination and pollen tube elongation of *Arabidopsis* [18]. Wang et al. pointed out that NO affects the formation of pollen tube cell walls by changing extracellular Ca^2+^ influx and F-actin organization, thereby affecting the elongation of *Pinus bungeana* pollen tubes [19]. These studies confirmed that NO can participate in the correct guidance of the pollen tube to the ovule, and it is mainly reflected in the regulation of the influx of extracellular Ca^2+^ and the organization of actin filaments in the process of cell wall construction. In addition, NO also participated in plant resistance to diverse abiotic stresses, for example, the high salinity [20], high temperature [21], drought [22] and LT [23]. Sehrawat et al. proposed that NO protects plants from LT damage by increasing the antioxidant defense of plants, preventing ROS damage [24]. Therefore, there may be a signal regulation network dependent on NO in the response of plants to LT. To explore the mechanism of NO signaling molecules’ involvement in the LT response of *C. sinensis* pollen tubes, we have initiated some studies and found that NO can positively regulate ROS and proline, aggravate the acidification and destroy the calcium ion concentration gradient of the pollen tube to deepen the effect of LT on the pollen tube [16,25,26]. However, further research is needed to clarify the underlying mechanism of this process.

MicroRNAs (miRNAs), a type of single strand, non-coding regulatory small RNAs (sRNAs), can modulate the expression level of the corresponding target genes through the cleavage or transcriptional inhibition of mRNA in plants [27]. In addition to the important role of miRNAs in cellular, biological processes and developmental functions [28], large number of studies have reported that a great deal of miRNAs play important roles in abiotic stresses induced by LT [29], heat [30], drought [31], salinity [32] and heavy metal [33]. In recent years, existing studies have explored the response of *C. sinensis* miRNAs to LT. The miRNA family, including miR156, miR164, miR166, miR167, miR171, miR398 and miR408, has been analyzed in the LT-response of *C. sinensis*, and the corresponding targets are shown to be involved in the synthesis of signaling pathways and cold stress response proteins [34,35]. Although the correlation between miRNAs and LT in different plant species has been confirmed, there is no research on the role of miRNAs in *C. sinensis* pollen tube growth in response to LT, especially the part in which NO participates in the response process. In our previous studies, some progress has been made in morphology, cytology, physiology and transcriptomics of NO participating in pollen tube response to LT [16,25,26]. Thus, the role of miRNA in response mechanism still needs to be further explored.

In the present study, we systematically identified the miRNA expression profiles in *C. sinensis* pollen tubes under LT and NO conditions for the first time. The function of miRNA in *C. sinensis* pollen tube response to LT and NO was studied by gene ontology (GO) and kyoto encyclopedia of genes and genomes (KEGG) pathway analysis. We believe that the present study will provide important data for understanding the roles of LT- and NO-response miRNA in regulating the diversified biological pathway in *C. sinensis*.

## 2. Materials and Methods

### 2.1. Pollen Source and Culture

We collected the mature pollen from *C. sinensis* cv. Longjingchangye on the day before blooms in November 2019. The *C. sinensis* plantation was located at Sun Yat-sen Tea Factory in Jiangsu Province, China (32°2′57″ N, 118°50′35″ E). The detailed information of in vitro pollen culture condition was mainly based on the previous publication [25]. Briefly, the pollen need be pre-incubated in the dark at 25 °C for 30 min in vitro with the standard culture medium. The pre-incubated pollen transferred to a 4 °C refrigerator and kept in the dark was regarded as the LT treatment. The pharmacological treatment was achieved via adding 25 µM NO donor DEA NONOate (NO). At the same time, the control (CK) treatment was carried out together. Unless otherwise specified, the following experiments were performed after 1 h of treatment. The pollen after germination (i.e., pollen tubes) was filtered with a nylon sieve with a pore size of 0.74 μm to remove the ungerminated pollen grains and culture medium, then frozen in liquid nitrogen and stored at –80 °C. Three biological replicates were performed for each treatment.

### 2.2. miRNA Library Construction and Sequencing

Total RNA was extracted from pollen tubes of three independent biological replicates for each treatment using RNAiso Plus (TaKaRa, Dalian, China, Cat NO. 9108) following the manufacturer’s protocol. In addition, the Fruit-mate for RNA Purification (TaKaRa, Dalian, China, Cat NO. 9182) was used to resolve high levels of polysaccharides and polyphenols in pollen tubes. The methods of the quality and integrity of total RNA and the constructing and sequencing cDNA library refer to Wang et al. [25].

### 2.3. Fundamental Analysis for miRNA

The raw reads obtained from small RNA sequencing were evaluated first, including filtering of impurities and preliminary determination. Impurities removed from raw reads generally include low quality reads, non-inserted fragment reads, long inserted fragment reads, polyA tails and small fragments reads. The procedures and parameters used in these processes are provided by a sequencing company (Beijing Genomics Institute, Shenzhen, China), and the pretreatment of miRNA is completed by BGI to obtain clean reads [36]. Secondly, the read types (represented by unique small RNA) and the number of reads (represented by total small RNA) of small RNA in clean reads were counted, and statistics on the length distribution of small RNA reads were performed. Rfam v.10.1 (http://www.sanger.ac.uk/software/Rfam) [37] and the GenBank database (http://www.ncbi.nlm.nih.gov/genbank/) [38] were used for BLAST alignment [39] to remove other small RNA and finally identify miRNA through hairpin structure.

Since there is no miRNA information on *C. sinensis* in the miRbase database, we first compared the clean data of the miRNA of the sample with the mature miRNA sequence of the plant in miRbase v.21 database (http://www.mirbase.org/) [40]. First, considering the difference among species, we aligned clean data to the miRNA precursor/mature miRNA of all plants in miRBase, allowing two mismatches and free gaps. Second, we chose the highest expression miRNA for each mature miRNA family which was regarded as a temporary miRNA database. Third, we aligned clean data to the above temporary miRNA database and the expression of miRNA was generated by summing the count of tags which could align to the temporary miRNA database within two mismatches. Finally, we predicted the precursor of the identified miRNA, of which unable to fold hairpin structure will be regarded as pseudo-miRNA. The feasibility of our result can be greatly improved by this verification. The above-mentioned construction process of miRNA expression profile known to *C. sinensis* is carried out by BGI using its self-developed software tag2miRNA. Additionally, the novel miRNAs were predicted by Mireap (http://sourceforge.net/projects/mireap/) [41]. The key conditions were as follows: the tags which were used to predict novel miRNA were from the unannotated tags which could match to reference genome, intron region and antisense exon region; those genes whose sequences and structures satisfied the two criteria, hairpin miRNAs that could fold secondary structures and mature miRNAs that were present in one arm of the hairpin precursors, would be considered as candidate miRNA genes; the mature miRNA strand and its complementary strand (miRNA*) presented 2-nucleotide 3′ overhangs; hairpin precursors lacked large internal loops or bulges; the secondary structures of the hairpins were steady, with the free energy of hybridization lower than or equal to −18 kcal mol^−1^; the number of mature miRNA with predicted hairpin must be no fewer than 5 in the alignment result.

### 2.4. Identification of Potential miRNA Target Genes and Annotation Analysis

The potential targets of the DEMs (include known miRNAs and novel miRNAs) were predicted by using the psRNATarget online server (http://plantgrn.noble.org/psRNATarget/) [42] based on the *C. sinensis* genomic coding sequences (http://tpia.teaplant.org/index.html) [43], with the default setting at maximum cutoff score based on a given scoring schema as 3, the seeds region between 2 and 7 nt, and the length for complementarity scoring (HSP size) at 19. In addition, bulge (gap) was allowed with a penalty for opening gap at 2 and penalty for extending gap at 1. The function (GO) and metabolism (KEGG) analysis of DEMs target genes were carried out by using the hypergeometric distribution method (ClusterProfiler software package, version 3.16.1, HK, China, 2020) of R software (R, version 4.0.2, Auckland, New Zealand, 2020) [44].

### 2.5. RT-qPCR Analysis of miRNAs and Its Targets

For RT-qPCR analysis of miRNA, first-strand cDNA was synthesized using the *TransScript*^®^ miRNA First-Stand cDNA Synthesis SuperMix (Transgen, Beijing, China, Cat No. AT351). RT-qPCR was conducted using the *TransStart*^®^ Top Green qPCR SuperMix (Transgen, Beijing, China, Cat No. AQ131) with the universal miRNA qPCR reverse primer supplied with the kit and miRNA specific forward primer (Appendix A). The 5.8s rRNA was selected as the internal control [45].

For RT-qPCR analysis of the targets, the HiScript^®^ III RT SuperMix for qPCR with gDNA wiper (Vazyme, Nanjing, China, Cat No. R323-01) was used to synthesize the first-strand cDNA. RT-qPCR analysis was conducted using the ChanQ^®^ SYBR qPCR Master Mix (Vazyme, Nanjing, China, Cat No. Q311-02) with the specific primer pairs showed in Appendix A. *β*-Actin served as a reference gene [46].

All the RT-qPCR was performed on the Bio-Rad BFX96 fluorescence (Bio-Rad C1000 Touch^TM^ Thermal Cycler). Each sample was run in three technical triplicates with three biological replicates. The specificity was confirmed by the melting-curve analysis of the amplified products at the end of the PCR. The expression level of each miRNA and target were normalized to the internal control genes based on the 2^−∆∆Ct^ method [47].

### 2.6. Statistical Analysis

Differentially expressed miRNAs (DEMs) were confirmed with the log2(Fold Change) > 1 or log2(Fold Change) < −1 and *p*-value < 0.05. The fold changes were calculated by the ratio of average miRNA expression in each treatment group and the p-values were calculated by T-test. Data analysis and correlation analysis were performed using SPSS software (SPSS Inc. version 22.0, IL, Chicago, USA, 2013). The network diagrams were drawn with Cytoscape software (Cytoscape, version 3.7.1, MD, Bethesda, USA, 2019), and other data diagrams were drawn with R software (R, version 4.0.2, Auckland, New Zealand, 2020).

## 3. Results

### 3.1. Small RNA Sequencing and Annotation of miRNAs from C. sinensis Pollen Tubes

Three RNA-seq libraries from CK, LT and NO treatments were constructed for high-throughput sequencing. >37.8 million total raw reads were generated with ~12.7 million reads from CK, ~12.4 million reads from LT and ~12.8 million reads from the NO (Appendix A). After filtering low quality reads and adapter trimming, ~12.4, ~12.2 and ~12.3 million clean reads were obtained that corresponded to ~3.84, ~3.80 and ~3.72 million unique reads in CK, LT and NO treatments, respectively [36]. About 7.82, 6.73 and 8.31% of the unique reads from respective libraries mapped to the *C. sinensis* genome (Appendix A). The size distribution of small RNA shows that the maximum length of read is 24-nt, followed by 21-nt (Appendix A).

The unique reads (18~24-nt) mapped to the genome (857,375) were further analyzed for miRNA identification. After removing the reads mapped to other non-coding RNAs and repetitive regions, the unique reads were identified by the method mentioned in part 2.3. A total of 13,278, 12,657 and 14,343 sequences were mapped to known miRNAs from CK, LT and NO libraries, respectively. This analysis resulted in a total of 284 unique conversed miRNA sequences (known miRNAs, Appendix A) Most notably, the highest expression level (~1,300,000) was detected for miR156c which exceeded most known miRNAs (Appendix A). The remaining ~0.73 million reads were used to predict novel miRNA. Through the prediction of the hairpin structure and the detection of both ends of the miRNA/miRNA* complex, a total of 186 miRNA candidates were considered novel or *C. sinensis*-specific miRNAs. Finally, combined together, 470 miRNAs (284 known and 186 novel) were identified from the *C. sinensis* pollen tubes under control, LT treatment and NO treatment (Appendix A). In addition, compared with known miRNAs, the expression level of novel miRNAs was shown to be quite reduced (Appendix A).

### 3.2. The Impact of Low-Temperature and NO on miRNA Expression

For known miRNAs, in total, 55 and 54 DEMs were identified in pollen tubes after LT and NO treatments, respectively. Among them, LT and NO treatment stimulated the expression of 25 and 16 miRNAs, and inhibited the expression of 30 and 38 miRNAs, respectively (Figure 1a and Figure 2a). For novel miRNAs, in total, 22 and 17 DEMs were identified in pollen tubes after LT and NO treatments, respectively. Among them, LT and NO treatment stimulated the expression of 9 and 5 miRNAs, and inhibited the expression of 13 and 12 miRNAs, respectively (Figure 1b and Figure 2b). It is worth noting that most of the 34 LT and NO co-induced DEMs showed the same expression pattern under LT and NO (Figure 1c,d).

### 3.3. Validation of Differentially Expressed miRNAs by qPCR

To verify the gene expression profile in our miRNA-Seq results in this study, the expression levels of 9 random DEMs after LT and NO treatments were verified using qRT- As showed in Figure 3, the relative expression level of the selected miRNA showed a consistent trend with the sequencing results, which proved the accuracy of the sequencing data (Figure 2). 

### 3.4. Prediction and Functional Analysis of Target Genes of miRNA

The current study used the cDNA sequences of 33,381 genes in the *C. sinensis* transcriptome data published on TPIA (http://tpia.teaplant.org/index.html) and the psRNA Target online server to identify miRNA targets. The analysis revealed that a total of 1616 *C. sinensis* genes were targeted by 253 miRNAs. The detail of the miRNAs and corresponding target genes is shown in Appendix A. A total of 3349 recognized miRNA target modules have been obtained with 3290 miRNAs that play a role through psRNA target-directed cleavage targets, and 59 miRNAs may play a role through translation inhibition. However, some miRNA targets were not predicted due to lack of genome information. Moreover, for LT-regulated-only DEMs, 69 miRNA target modules were screened involved 66 target genes; for NO-regulated-only DEMs, 264 miRNA target modules were screened and it was shown that 246 target genes were involved; as for LT and NO co-regulated DEMs, 33 unique miRNA target modules were screened (Appendix A).

To further comprehend the functional significance of these modules and determine the most relevant targets for cold-induced and NO-induced in-depth identification, gene ontology (GO) and KEGG pathway enrichment analysis on target genes corresponding to DEMs were performed.

The results showed that 42 targets were annotated with 86 GO terms that were enriched under LT (Appendix A), including 25 biological processes (BP), 24 cellular components (CC) and 37 molecular functions (MF). Most of these GO terms were associated with abiotic stress tolerance and pollen tube growth, for instance, cell wall organization carbohydrate metabolic process, microtubule-based movement, external encapsulating structure organization, microtubule-based process and vesicle-mediated transport in BP; and oxidoreductase activity, actin filament binding, cellulose synthase activity, cytoskeletal protein binding, motor activity, microtubule binding in MF. The target genes enriched in most of these GO terms were targeted by miR5819, miR9563b-3p, miR1535a, miR1873, miR5721 and novel_mir_18 (Figure 4).

As for NO treatment, a total of 115 targets were annotated with 196 GO terms including 73 biological processes, 33 cellular components and 90 molecular functions (Appendix A). NO donor regulates the activation of more genes and a variety of biological processes and makes more cellular components participate in the growth of *C. sinensis* pollen tubes. As compared with LT, exogenous NO regulated cation binding and oxidation-reduction process, in particular, which included potassium channel activity, calmodulin binding, copper ion binding, magnesium ion binding, peroxidase activity, antioxidant activity, cellular oxidant detoxification, oxidation-reduction process and response to oxidative stress. The target genes of miR396g-3p, miR172c-3p, novel_mir_18 and novel_mir_37 were enriched in most of these GO terms (Figure 5). In particular, the target gene of novel_mir_37 exhibited powerful redox-related functions.

The GO modules were also divided into ‘LT-regulated-only’ and ‘LT and NO co-regulated’ according to the regulation mode of miRNAs to further explore the function of NO in the *C. sinensis* pollen tube response to LT. The results indicated that the targets of DEMs that are responsive to LT can be located on a variety of cellular components, while the target genes of miRNAs co-regulated by LT and NO are only located on the membrane and wall around the cell. Moreover, microtubule-related and actin-related GO functions were not found in ‘LT and NO co-regulated’. In Figure 6, the target genes of co-regulated miRNA significantly enriched on the protein-cysteine S-palmitoyl transferase activity, potassium ion transmembrane transporter activity, external encapsulating structure, enzyme regulator activity, cell wall organization and peptidase activity, acting on L-amino acid peptides. The target genes of novel_mir_18 points to most of the GO terms. Additionally, the integral component of membrane and metal ion binding was enriched by the target genes of multiple DEMs.

The target genes of DEMs for LT/CK and NO/CK pointed to 58 and 73 KEGG pathways, respectively (Appendix A). It must be noted that 49 pathways were all enriched in both comparisons, mainly including alanine, aspartate and glutamate metabolism, AMPK signaling pathway, apoptosis, flavonoid biosynthesis, MAPK signaling pathway, starch and sucrose metabolism, valine, leucine and isoleucine biosynthesis. Regulation of actin cytoskeleton was specifically enriched only in LT/CK (Appendix A). In NO/CK, the targets of DEMs were more enriched in the metabolic pathways of carbohydrates and amino acids (Appendix A). In LT treatment, biosynthesis of secondary metabolites and mRNA surveillance pathway were the two main pathways (Figure 7); therein, the targets of miR1535a, miR3515, miR845b, miR5819, miR1873 and novel_mir_23 participated in the synthesis and metabolism of various secondary metabolites. As for NO treatment, the targets of miR172c-3p, miR5293 and novel_mir_37 were involved in saccharo metabolism and biosynthesis of amino acids (Figure 8).

### 3.5. Expression Analysis of miRNA and the Targets by qPCR

To further explore and verify that NO signaling molecules can participate in *C. sinensis* pollen tube response to LT, the relative expression of 8 essential DEMs and the corresponding targets were chosen for qRT-PCR validation after being treated with LT, 25 µM DEA NONOate (NO) and 200 µM 2-(4-carboxyphenyl)-4,4,5,5-tetramethylimidazoline-1-oxyl-3-oxide (cPTIO, a NO scavenger) under 4 ℃ (LT+cPTIO). Nine of these target genes showed miRNA-related expression patterns and encoded the following: terpene synthase (TEA031966.1), phospholipase A2-alpha-like (TEA030352.1), protease-associated RING/U-box zinc finger family protein (TEA018965.1), potassium transporter (TEA022010.1), S-acyltransferase 19 isoform X1 (TEA000142.1), glycogen phosphorylase 1-like (TEA030640.1), adaptor protein complex AP-1, gamma subunit (TEA019788.1) and DnaJ protein homolog (TEA015484.1). As shown in Figure 9, with the addition of cPTIO, the changes in the expression levels of LT-induced miRNA and the corresponding targets were relieved, indicating that these miRNAs might be involved in the response of *C. sinensis* pollen tubes to LT by regulating the NO signal pathway. Additionally, 8 miRNA-target modules displayed an inverse correlation, whereas miR1873-TEA019788.1 exhibited positive correlation with higher expression of both miRNA and target under LT (Figure 9).

## 4. Discussion

The negative effects of LT on pollen tubes have been confirmed in many plants including *C. sinensis* [25]. Studies in recent decades have shown that NO plays a vital role in plant resistance to various abiotic stresses [48]. According to our previous studies, exogenous cPTIO can effectively reduce the damage of cell wall ultrastructure caused by LT by slowing down the acidification of pollen tube tip and ROS accumulation [25]. In addition, comprehensive gene expression profile analysis revealed the possible molecular mechanism of NO mediated the growth of *C. sinensis* pollen tube under LT [26]. These results provide basic cytological and transcriptional resources for further exploring the molecular mechanism of high temperature resistance in *C. sinensis* reproductive tissues. To further clarify the molecular basis of abiotic stress affecting the growth of *C. sinensis* pollen tubes, the miRNA omics analysis was performed in the present study. The results showed that 332, 323 and 321 miRNAs were identified from the CK, LT and NO, respectively. Among them, miR156c reported the highest numbers. In addition, we found that the expression level of novel miRNAs is much lower than the known conserved miRNAs, which was similar to the previous *C. sinensis* miRNA sequencing results [35].

In current study, in total, 77 DEMs were confirmed in pollen tubes under LT, with 71 under NO (Figure 1). miR1077-5p, miR6170, miR1429-3p, miR3515, miR9667-5p, miR407, miR2590h, miR845b and miR845c have the highest expression level under the treatment of LT which may regulate the expression of the key cold-resistant gene. However, the common LT-responsive miRNAs identified in studies on other plants subjected to LT did not show significant differences in this study, such as miR156, miR167, miR172, miR396, miR398, miR408 in *Musa* [30], *Zea mays* [49], *Astragalus membranaceus* [29] and even in *C. sinensis* [34,35]. This seems to give us a hint that the miRNA-related molecular mechanisms of plant growth and reproductive development in response to LT environments are different. This phenomenon is also found in other species, such as rice and wheat. Jiang et al. found that miR166, miR395, miR396, miR408, miR1425 and miR1861c are the main DEMs in two and half-leaf stage rice seedlings under cold treatment [50], while Maeda et al. pointed out that the main DEMs in rice anthers after cool-temperature stress are miR397b, miR398b, miR408-3p, miR171b/c/d/e/f, miR528-5p, miR1432-5p and miR5072 [51]. Similarly, LT induced the differential expression of miR159, miR164, miR168, miR172, miR393, miR397, miR529, miR1029 in two-week-old wheat seedlings [52], and that of miR9669-5p, miR397-5p, miR9658-3p, and miR9672b in wheat microspores [53]. Although there may be specific expression miRNAs caused by genotypes, these findings seem to provide evidence that plant growth and reproductive development have different miRNA-related molecular mechanisms in response to LT treatment. Moreover, 34 LT- and NO-induced co-regulatory miRNAs were identified (Figure 1) which seem to give us a hint that NO and LT treatments have a synergistic effect on the restraint of *C. sinensis* pollen tube development. The performance of this negative regulation has been confirmed in our previous work [25]. Therefore, these results suggest that the co-regulated miRNAs seem to be a potential molecular mechanism for NO to participate in the process of LT affecting the growth of *C. sinensis* pollen tubes.

Meanwhile, other studies have also found the LT- or NO- induced miRNAs during the development of pollen tubes identified in this study. The gma-miR2118a/b from the miR2118 family was found to be highly expressed in soybean shoot apex meristems (SAM), and may promote the polar growth of SAM by down-regulating their target protein expression in SAM [54]. Similarly, the down-regulation of csn-miR2118a-3p under LT and NO treatment may be one of the reasons for inhibiting the polar growth of *C. sinensis* pollen tubes. In addition, miR2118 leaves were found to be related to abiotic and biotic stress in soybeans [55]. Therefore, these DEMs identified in this study provide potential candidate genes for the analysis of miRNA target modules that regulate pollen tube growth in response to LT treatment and NO treatment. Furthermore, we found many miRNAs from the same miRNA families showed a different regulatory pattern in response to LT (miR845 and miR2118) and NO (miR167, miR172 and miR2118) treatment (Figure 2a). Such results correspond to findings in previous studies, and the most likely explanation is that their target mRNAs are diverse, which leads to different functions of miRNAs within the same family [34,35]. 

The elongation of the pollen tube and its normal shape involve a series of complex cell activities, including reasonable ion flux, proper deposition of cell wall saccharides and organization of actin [11,56]. In the current study, LT-induced miRNAs, miR5819, miR9563b-3p, miR1873 and miR5721 targeted actin filament/cytoskeleton related mRNAs (Figure 4, Table 1), indicating that these miRNAs most likely affected the normal organization of *C. sinensis* pollen tube actin through the interaction with their target genes. As the most important factor to maintain pollen tube morphology, actin filaments can cooperate with other processes such as Ca^2+^ flux to effectively make vesicles gather at the tip of the pollen tube and fuse with the plasma membrane. However, when these factors are unstable due to any external environment, the vesicles will fuse in a larger area of plasma membrane except the tip of the pollen tube, which will lead to abnormal development of the pollen tube [11]. TEA010276.1, a target gene of miR5819, is coding for motor protein, a 125 kDa kinesin-related protein, which represents the key to a variety of cell activities, such as movement, division, transportation and maintaining cell shape and mechanical integrity [57]. The correct positioning of organelles and molecules in space and time is critical to their function, and motor proteins are thought to participate in their local positioning by promoting short-distance movement of organelles and molecules along microtubules [57,58]. The target gene of miR5721, TEA013890.1, encodes an actin binding protein, villin (Appendix A), which can directly regulate the actin dynamics [59]. The actin binding protein (APB) can transform the stimulations of the cell into the changes of the actin architecture and mediate the actin cytoskeleton rearrangement under various stimuli [60]. In some previous reports, *Arabidopsis* villins had a complete set of actin-related regulatory activities, including the assembly of nucleated actin and the binding, covering and cutting of actin filaments [61,62,63]. Villin is a Ca^2+^ reactive ABPs and responds to the increase of Ca^2+^ by mediating the change of actin in the pollen tube [64]. In the study of Zhao et al., the elevatory Ca^2+^ caused villins to sever the actin filaments in the pollen tube, and at the same time villins aggregated the fragmented actin filaments into actin foci and expanded them [65]. In addition, under low Ca^2+^ conditions, villin has been proven to stabilize actin filaments by diluting- and profilin-mediated actin depolymerization [63]. Therefore, these microtubules/actin filament-related DEMs induced by LT essentially affect the growth of *C. sinensis* pollen tubes.

LT can easily inhibit pollen tube growth by affecting cell membrane and cell wall changes. LT is sensed at the membrane level and alters their fluidity. This sensory mechanism triggers a cascade of intracellular signals that leads to a cellular response [3]. When pollen cells are exposed to LT, a cascade of intracellular signals is triggered, leading to the production of reactive oxygen species (ROS). Excessive ROS can cause damage to cells and destroy the integrity of cell membranes [66,67]. Cytochrome P450 genes (CYPs) are one of the largest gene families in plants, involved in various biological processes such as biotic and abiotic stress responses [68]. In higher plants, CYPs, as a multifunctional catalyst, play an important role in the biosynthesis of a large number of compounds and metabolites, such as antioxidants, plant hormones, structural polymers and signal molecules [69]. In this study, the target gene TEA026834.1 of miR9563b-3p seems to be a CYPs and encodes Cytochrome P450 71D11-like. The up-regulation of miR9563b-3p induced by LT may lead to the down-regulation of its target gene TEA026834.1, thereby weakening the plant’s catalytic effect on antioxidant synthesis and ultimately leading to the accumulation of ROS. Similarly, Wang et al. found that overexpression of *TaCYP81D5* accelerated the elimination of active oxygen in wheat under salt stress [68]. Furthermore, cytochrome P450 has also been reported as a NO biosynthetic enzyme [70], and 4 CYPs co-regulated by LT and NO were also screened in our previous transcriptome analysis [25]. However, in this study, although most mRNAs related to redox process are the target genes of novel_mir_37 which induced by NO treatment, none of them encode CYPs (Figure 5). These results indicate that NO may not participate in the response of the *C. sinensis* pollen tube to LT through redox process, or there is another possibility that NO can participate in the response of the *C. sinensis* pollen tube to LT through redox process, but there is no miRNA regulation.

In addition, a novel miRNA, novel_mir_18, predicted in this study, which is specifically expressed only in the control, attracted our attention. The target genes of novel_mir_18 we predicted point to most of the functions significantly enriched by co-regulation of miRNA targets, as well as potassium ion transmembrane transporter activity and plasma membrane (Figure 6, Table 1). Through further study of its target genes, it was found that both TEA020146.1 and TEA028568.1 encode pectinesterase 2 and are enriched in external encapsulating structure and cell wall organization. Pectin esterase has been confirmed to participate in the plasticity of the apical cell wall of the pollen tube by regulating the deesterification of methyl esterified pectin [71]. The cell wall of the pollen tube is uneven, because the composition of its growth axis is different, and it has the characteristics of pectin layer, which extends to the whole tube length. At the shank of the pollen tube, the pectin layer is gradually strengthened by the cell wall inner layer mainly containing callose and cellulose [2,3]. Previous studies reported that LT caused the cell walls of maize and eucalyptus to thicken and increased the expression of genes related to lignin deposition [4,5]. This unusual and uneven lignification of the pollen tube apex cell wall slows its growth and may result in abnormal growth. In addition, novel_mir_18 may also influence growth and development by regulating metabolism. We have noticed that TEA020146.1 and TEA028568.1, the targets of novel_mir_18, which encode pectinesterase 2, took part in the carbohydrate metabolism via pentose and glucuronate interconversions. Moreover, another novel_mir_18 target gene, TEA023416.1, pointed to the GO function hydrolase activity, hydrolyzing O-glycosyl compounds and carbohydrate metabolic process by encoding glucan endo-1,3-beta-glucosidase 8 (Figure 4). The amazing growth rate of pollen tubes and the extreme polar growth of single cells need to absorb a large amount of carbohydrates from the outside to consume energy and provide biosynthesis of new cell walls [72,73]. Any disturbance of carbohydrate supply or transportation capacity may seriously damage the development of pollen or the elongation of pollen tubes. LT and NO ultimately lead to abnormal pollen tube growth mediated by glucose metabolism disorder through the regulation of miRNA. Moreover, apoplastic pH and the availability of divalent cations could regulate pectin esterase to control cell wall stiffening and cell wall loosening [74,75]. Previous studies have found that NO signal can transport monovalent and divalent cations through cGMP-activated and/or cAMP-activated cyclic nucleotide-gated ion channels (CNGC) [25,76]. In addition, our previous study has identified two glutamate receptor-like channels genes from differentially expressed genes of LT/CK and NO/CK in *C. sinensis* pollen tubes which were identified as a putative group of pollen Ca^2+^ channels with the Ca^2+^ transport activities in pollen tubes [25,77,78]. Therefore, we boldly proposed a possible signal network which implies that NO participates in regulation of pollen tube response to LT. According to this model, LT and NO change the Ca^2+^ gradient in the pollen tube by regulating GRLs, thereby affecting the expression of novel_mir_18 and regulating the translation of pectin esterase by targeting TEA020146.1 and TEA028568.1, and finally, leading to the abnormal growth of the *C. sinensis* pollen tube apex via regulating the pollen tube cell wall stiffening and loosening. Furthermore, the target gene TEA022010.1 of novel_mir_18 located on the plasma membrane encodes a potassium transporter, and the study of Hu et al. also showed that potassium deficiency can inhibit the growth of cotton pollen tubes in the style, which indicated that the NO signaling molecule might also regulate the response of *C. sinensis* pollen tubes to LT by mediating K^+^ transport [79]. The above results indicate that novel_mir_18 is very likely to be an important regulation method of the NO signaling molecule to further regulate the response of *C. sinensis* pollen tubes to LT. 

The results of RT-qPCR indicate that the target genes of miRNA co-regulated by LT and NO are related to the binding or transport of metal ions (Figure 9). Pollen germination and pollen tube elongation are mainly cell swelling reactions caused by cell water absorption, which is influenced by the intracellular and extracellular ionic environment [80], especially the Ca^2+^. We observed that TEA031966.1, TEA030352.1 and TEA018965.1, the target genes of miRNA down-regulated by LT and NO (miR1429-3p, miR1535a and miR4416b), all showed down-regulated expression under LT+cPTIO treatment compared with LT and NO treatments (Figure 9), designating their role in regulating the homeostasis of metal ions in pollen tube cells. As an important metal ion in pollen tube development, it is important to maintain the concentration gradient of Ca^2+^ at the tip of the pollen tube. LT and NO treatments may interrupt the concentration gradient of Ca^2+^ at the tip of the pollen tube by regulating these miRNAs, which indicates that NO signal molecules can participate in the response of the pollen tube to LT by regulating the concentration gradient of Ca^2+^ at the tip of the pollen tube through miRNA. Mitogen-activated protein kinases (MAPKs) have been proven to mediate the guiding reaction in pollen tubes by conducting upstream Ca^2+^ signals, and more and more studies have shown that MAPKs can participate in the apical growth of the pollen tube as intracellular target of NO [15,70,81]. As expected, some miRNAs, whose target genes participated in the MAPKs signaling pathway, were induced by LT and NO (Figure 7, Figure 8 and Appendix A). However, these miRNAs are all induced by LT or NO alone, and even the target genes enriched in the MAPK signaling pathway are not repeated. It is worth noting that most of these target genes participate in the MAPK signaling pathway (ko04010) by encoding interleukin-1 receptor-associated kinase 4 (IRAK4). However, at present, most of the research on IRAK4 focuses on animal immunity, while the research on IRAK4 in plants is still scarce. The few studies on IRAK4 in plants only suggest that it may be related to tiller development of *Festuca arundinacea Schreb* [82].

To sum up, the omics analysis of miRNA provides a more detailed network of the role of NO participation in the response of polarized tip growth *C. sinensis* pollen tubes to LT. A complex signal regulation network mainly relying on miRNA mediated by NO, including metal ion binding/transport, ROS, actin, carbohydrate metabolism and their crosstalk, is shown in Figure 10. The abnormal growth of the *C. sinensis* pollen tube caused by NO induced by LT is probably due to the binding and transport of metal ions mediated by miRNA and the change of Ca^2+^ flux mediated by various Ca^2+^ channels. Differential expression of miR5721, miR5819, miR1429-3p and novel_mir_18 regulated their target genes related to pollen tube growth, resulting in changes in insufficient energy, cell wall organization and ion distribution and for pollen tube growth, thus inhibiting polar growth of *C. sinensis* pollen tube tips after LT treatment. In summary, this study provides a new perspective for the mechanism of NO in the response of *C. sinensis* pollen tube growth to LT.

## Figures and Tables

**Figure 1 biomolecules-11-00930-f001:**
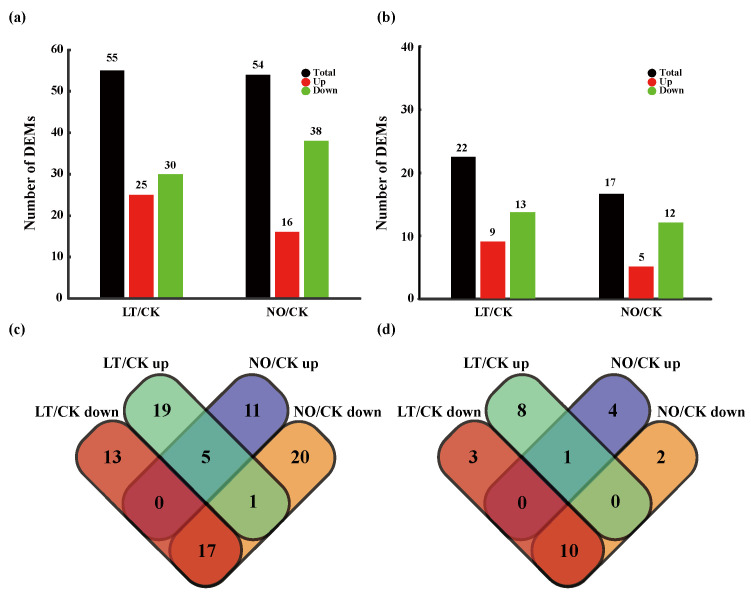
Summary of differentially expressed microRNAs (DEMs). Numbers of known DEMs (**a**) and novel DEMs (**b**) between LT (low temperature) and CK (control), and between NO (nitric oxide) and CK; Venn diagrams of known DEMs (**c**) and novel DEMs (**d**) between LT and CK, and between NO and CK. ‘LT/CK up’ and ‘LT/CK up’ represent the up-regulated and down-regulated DEMs between LT and CK, and ‘NO/CK up’ and ‘NO/CK up’ represent the up-regulated and down-regulated DEMs between NO and CK, respectively. The numbers in each cell indicate the number of miRNAs regulated by LT or NO alone, or co-regulated by LT and NO.

**Figure 2 biomolecules-11-00930-f002:**
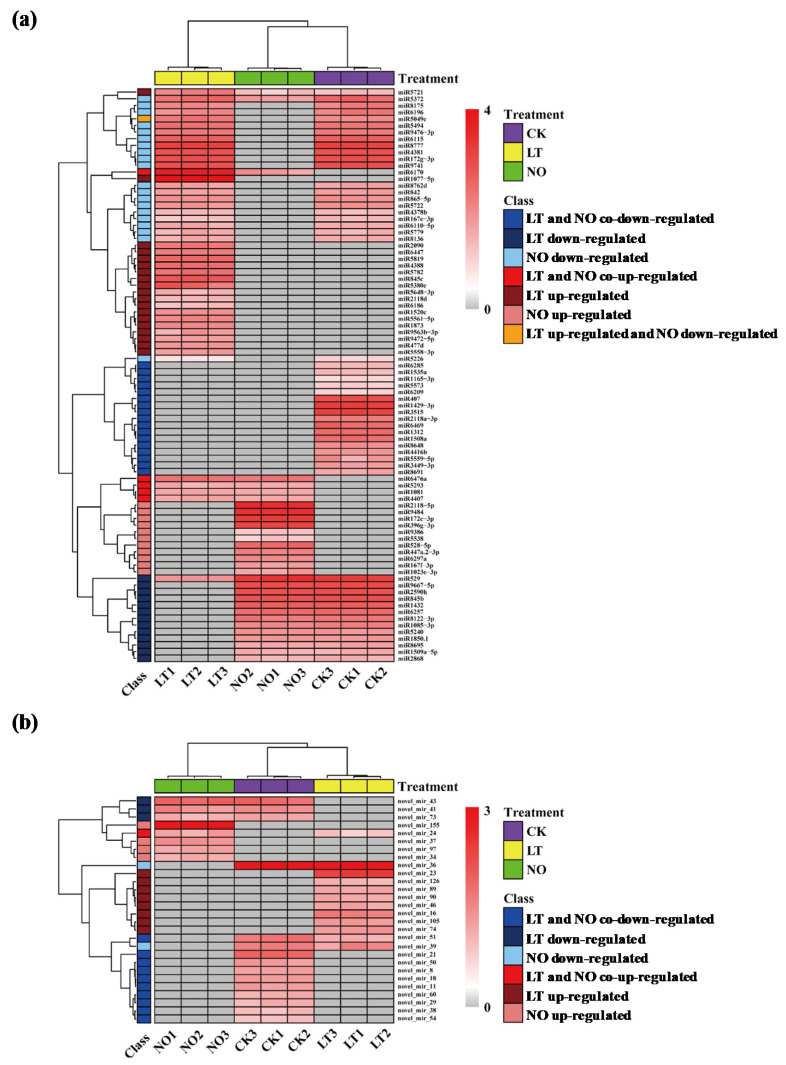
Hierarchical clustering analysis of known microRNAs (DEMs) (**a**) and novel miRNAs (**b**) expression in CK (control), LT (low-temperature) and NO (nitric oxide). CK1, CK2 and CK3; LT1, LT2 and LT3; NO1, NO2 and NO3 represent the three independent biological repetitions in CK, LT and NO, respectively. The gray square represents that the miRNA is not expressed in the sample. The cell colors from white to red represent the expression level of miRNA from low to high, and the value representing the cell color is the expression level of miRNA.

**Figure 3 biomolecules-11-00930-f003:**
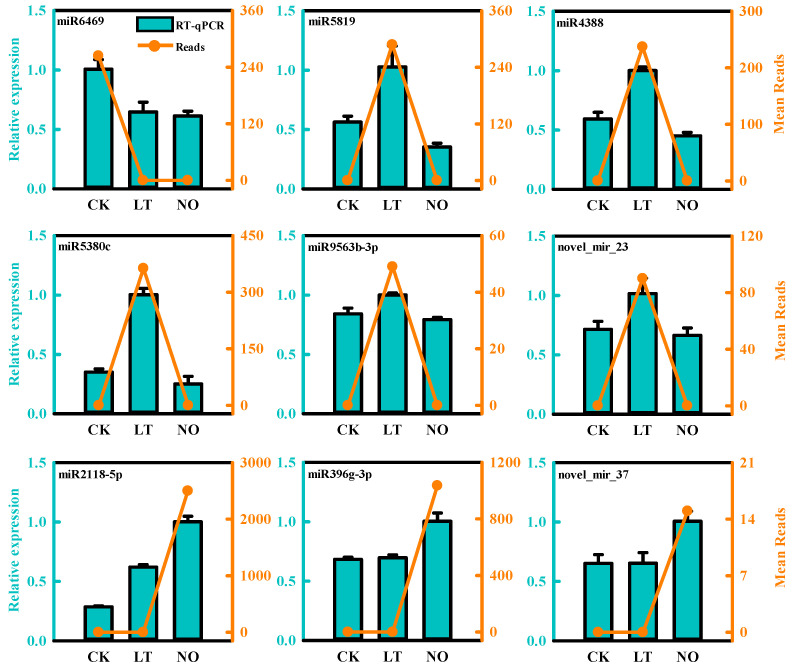
Expression validation of selected microRNAs (DEMs) in CK (control), LT (low-temperature) and NO (nitric oxide) by RT-qPCR. The error bars show standard deviation between biological replicates.

**Figure 4 biomolecules-11-00930-f004:**
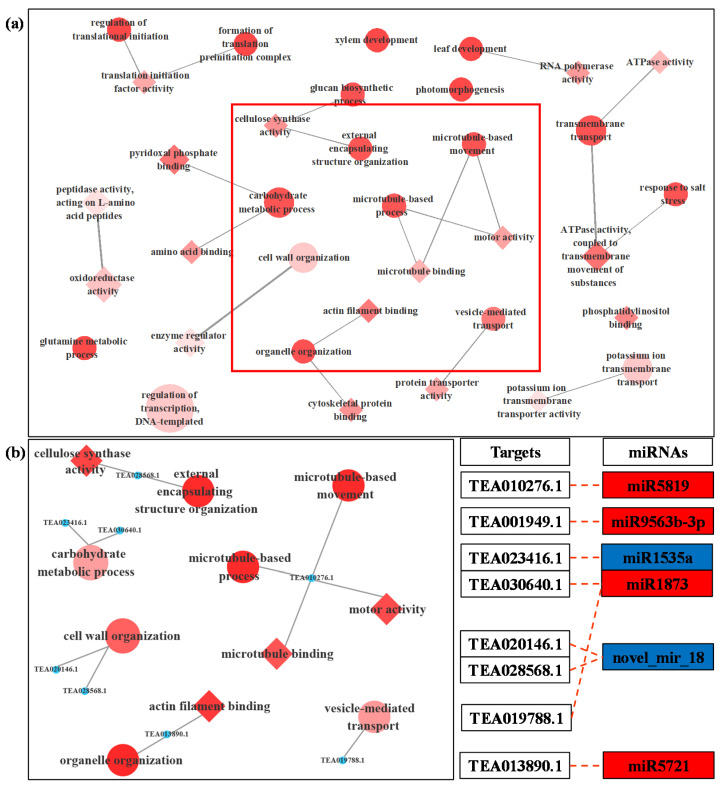
GO-GO network of the targeted genes of DEMs between LT (low-temperature) and CK (control) (**a**). Local amplification of GO-GO network related to microtubule and cell wall organization and its miRNA target (**b**). In the network, the circle represents the biological process term, and the diamond represents the molecular function term. The filling color of the GO term from white to red represents the enrichment degree (p.adjust) of the GO term. The color of the miRNA term indicates its regulation mode; red represents that miRNA is up-regulated in LT, and blue represents that miRNA is down-regulated in LT.

**Figure 5 biomolecules-11-00930-f005:**
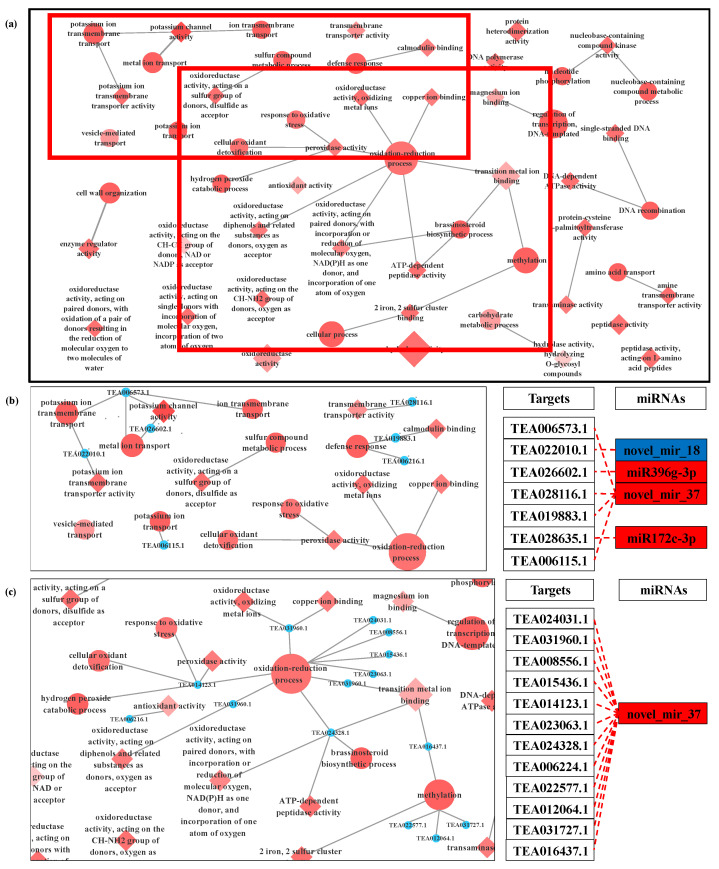
GO-GO network of the targeted genes of DEMs between NO (nitric oxide) and CK (control) (**a**). Local amplification of GO-GO network related to metal ion transport and its miRNA target (**b**). Local amplification of GO-GO network related to oxidation-reduction process and its miRNA target (**c**). In the network, circle represents the biological process term, and the diamond represents the molecular function term. The filling color of the GO term from white to red represents the enrichment degree (p.adjust) of the GO term. The color of the miRNA term indicates its regulation mode; red represents that miRNA is up-regulated in NO, and blue represents that miRNA is down-regulated in NO.

**Figure 6 biomolecules-11-00930-f006:**
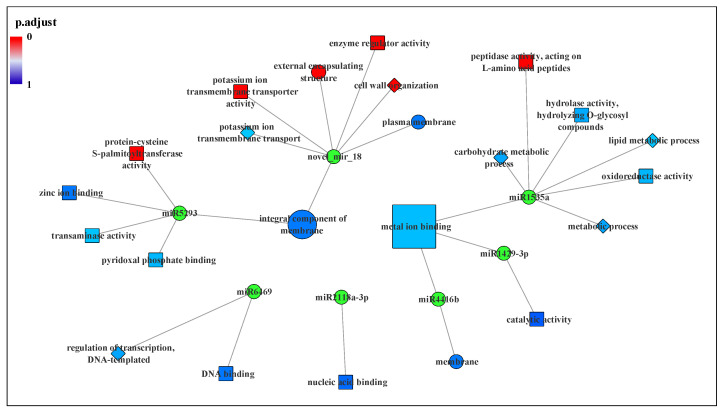
GO terms of target genes of miRNAs co-regulated by LT and NO. The circle represents the biological process term, the square represents the cellular components term and the diamond represents the molecular function term. The filling color of the GO term from blue to red represents the enrichment degree (p.adjust) of the GO term.

**Figure 7 biomolecules-11-00930-f007:**
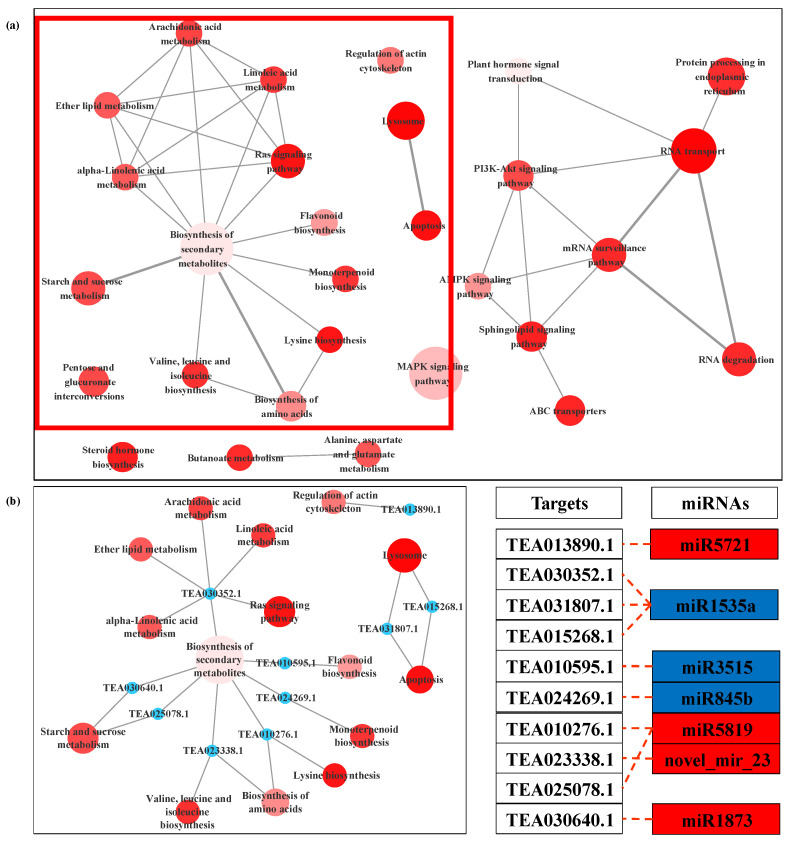
KEGG-KEGG network of the targeted genes of DEMs between LT (low temperature) and CK (control) (**a**). Local amplification of KEGG-KEGG network related to biosynthesis of secondary metabolites and its miRNA target (**b**). The filling color of the KEGG term from white to red represents the enrichment degree (p.adjust) of the KEGG term. The color of miRNA term indicates its regulation mode; red represents that miRNA is up-regulated in LT, and blue represents that miRNA is down-regulated in LT.

**Figure 8 biomolecules-11-00930-f008:**
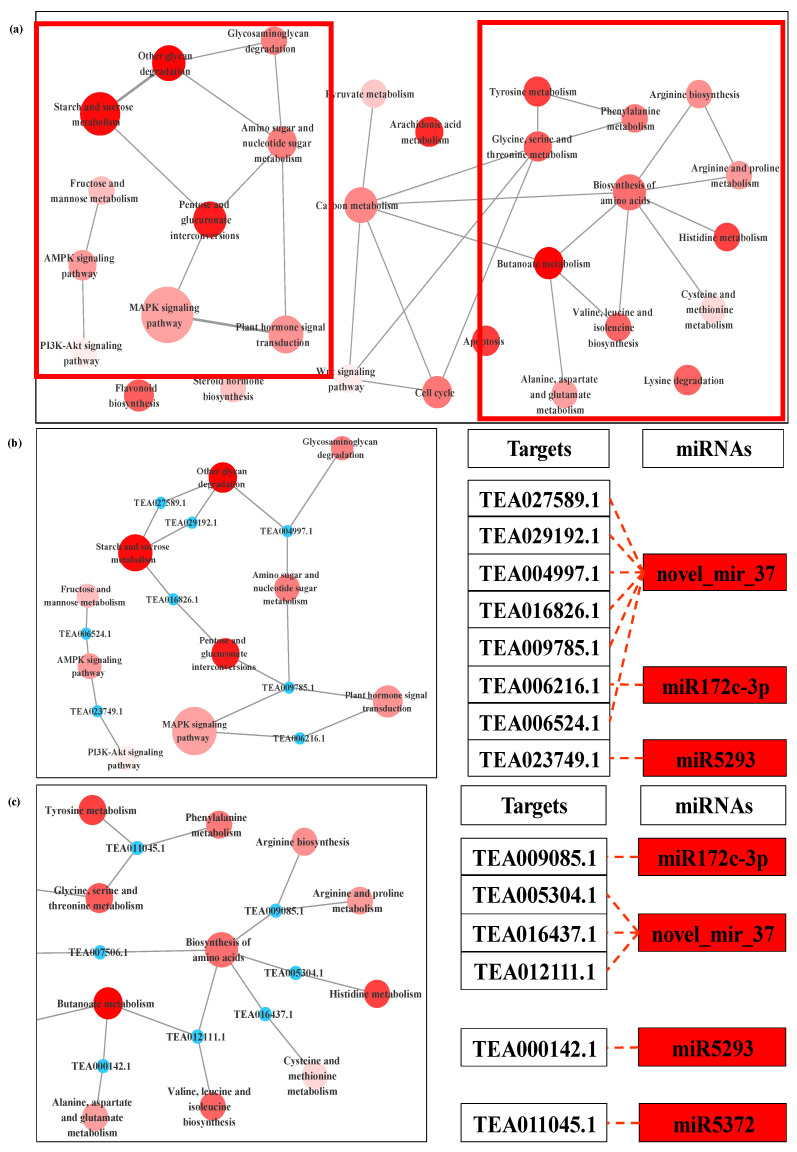
KEGG-KEGG network of the targeted genes of DEMs between NO (nitric oxide) and CK (control) (**a**). Local amplification of KEGG-KEGG network related to starch and sucrose metabolism and its miRNA target (**b**). Local amplification of KEGG-KEGG network related to biosynthesis of amino acids and its miRNA target (**c**). The filling color of the KEGG term from white to red represents the enrichment degree (p.adjust) of the KEGG term. Red miRNA term represents that miRNA is up-regulated in NO, and blue represents that miRNA is down-regulated in NO.

**Figure 9 biomolecules-11-00930-f009:**
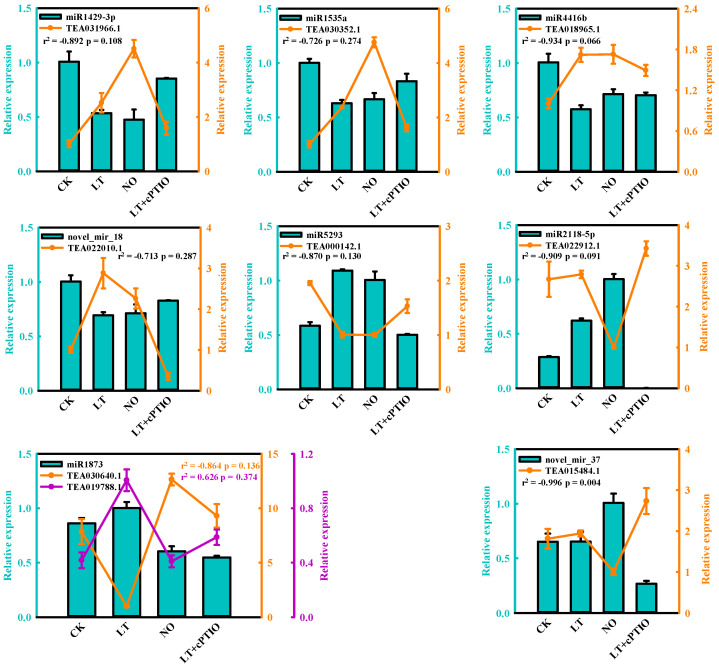
Expression validation of selected microRNAs (DEMs) and the target genes in CK (control), LT (low-temperature) and NO (nitric oxide) by RT-qPCR. The error bars show standard deviation between biological replicates. r^2^ represents the Pearson’s correlation factor.

**Figure 10 biomolecules-11-00930-f010:**
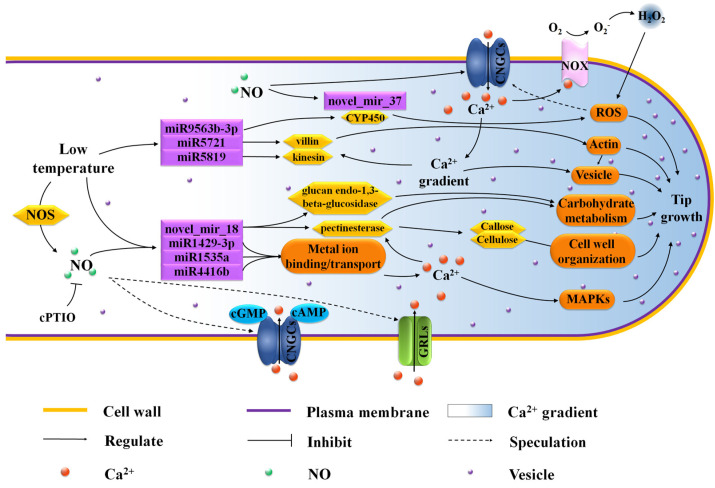
Potential hypothesis model that nitric oxide (NO) participates in low temperature inhibition of *C. sinensis* pollen tube growth by regulating miRNA. Low temperature further regulates the actin structure in tea pollen tube by mediating the expression of miR5721 and miR5819. The miR1429-3p and novel_mir_18 co-regulated by low temperature and NO can participate in the deposition of callose and cellulose, and the metabolism of carbohydrate through glucaendo-1,3-beta-glucosidease and pectinesterase. The above two regulatory pathways regulate the growth of the tea pollen tube from the aspects of component distribution, cell structure and energy supply, and are all affected by cytoplasmic Ca^2+^ gradient. Cytoplasmic Ca^2+^ gradient is mainly regulated by Ca^2+^ flux, which depends on various Ca^2+^ channels, such as CNGCs (cGMP-activated and cAMP-activated channels) and GLRs. To sum up, a complex signal network dominated by NO mediates pollen tube growth inhibited by low temperature. CNGCs, cyclic nucleotide-gated ion channels; cPTIO, 2-(4-carboxyphenyl)-4,4,5,5-tetramethylimidazoline-1-oxyl-3-oxide; CYP450, Cytochrome P450; GLRs, glutamate receptor-like channels; NOS, nitric oxide synthase; NOX, NADPH oxidase; ROS, reactive oxygen species.

**Table 1 biomolecules-11-00930-t001:** Co-regulated DEMs in LT/CK and NO/CK involved in NO signaling pathway under low-temperature.

miRNA	Up/Down Regulation	Potential Target Gene	Function Annotation
miR1312	Down	TEA029389.1	Protein TRANSPARENT TESTA 12
miR1429-3p	Down	TEA031966.1	Terpene synthase
miR1508a	Down	TEA027146.1	Peptide-N(4)-(N-acetyl-beta-glucosaminyl)asparagine amidase isoform X2
miR1535a	Down	TEA005439.1	BAG family molecular chaperone regulator 2
		TEA023416.1	Glucan endo-1,3-beta-glucosidase 8
		TEA030352.1	Phospholipase A2-alpha-like
		TEA005524.1	Regulator of chromosome condensation repeat-containing protein isoform 1
miR2118a-3p	Down	TEA017334.1	Cancer-related nucleoside-triphosphatase homolog isoform X2
		TEA028604.1	Extensin-2-like
		TEA016266.1	Polyadenylate-binding protein 8-like
miR3515	Down	TEA010595.1	Protein SRG1
miR4416b	Down	TEA018965.1	Protease-associated RING/U-box zinc finger family protein
miR5293	Up	TEA000142.1	S-acyltransferase 19 isoform X1
		TEA023749.1	Serine/threonine protein phosphatase 2A 55 kDa regulatory subunit B beta isoform
miR6469	Down	TEA023847.1	Dof zinc finger protein DOF5.6-like
		TEA001762.1	Hypothetical protein VITISV_027707
		TEA018038.1	Putative membrane protein insertion efficiency factor isoform X2
novel_mir_18	Down	TEA020146.1	Pectinesterase 2
		TEA028568.1	Pectinesterase 2
		TEA022010.1	Potassium transporter 2
		TEA004684.1	Transport protein SEC13 homolog B

## Data Availability

All data generated or analysed during this study are included in this published article [and its Appendix A].

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
