# Peer review of "MicroRNA Omics Analysis of Camellia sinesis Pollen Tubes in Response to Low-Temperature and Nitric Oxide"

_biomolecules, 2021, doi:10.3390/biom11070930_

Round 1
Reviewer 1 Report
This work describes an investigation of the differences in expression of microRNA as well as their targets in response to low-temperature and nitric oxide in Camellia sinensis pollen tubes after low-temperature and nitric oxide treatments using a transcriptome approach. Expansive analysis was performed confirming previous results and comprehending significant novel results providing a more detailed insight in the mechanism of nitric oxide in the response of C. sinensis pollen tube growth to low-temperature conditions.
Overall, the manuscript is very well written, with complete and corresponding Discussion section, leading to conclusions with emphasis on the significance of the accomplished results. However, I have several remarks, of which the first is regarded to the article title which is quite complex, and I suggest changing it to “MicroRNA omics analysis of Camellia sinesis pollen tubes in response to low-temperature and nitric oxide”. Furthermore, additional clarification needs to be provided regarding RT-qPCR analysis as I pointed out in my comments. Also, some of the sentences are long and not clear and concise, so they need to be rewritten in order to appropriately explain the presented results.
My major remark is concerning the presentation of the results since there are so many of them included in actually 10 figures along with supplementary material. My opinion is that the figure legends don't quite represent the complexity of the figures themselves. Hence, completing them with providing some additional information in figure description, i.e. image marks, symbol clarification, color/shape scale/meaning..... would be highly beneficial for better understanding of significant results.

Reviewer 2 Report
This paper reports the identification of differentially expressed miRNAs in low-temperature and nitric oxide exposure vs control in Camellia sinensis. The research follows a logical flow, going from the identification of differentially expressed miRNAs to the characterization of its targets by means of KEGG and GO term enrichment analysis founding some interesting genes. A validation with qRT-PCR it’s also included. However, some issues should be addressed before the manuscript is suitable for publication. My detailed comments are as follows:
Major concerns:
- Fundamental analysis for miRNA section:
- The methodology of miRNA pre-processing and analysis must be further described. No parameters and software it’s cited in the pre-processing steps and also in the alignment.
- C. sinensis it’s not included in miRbase, therefore the authors claim the following: “Since there is no miRNA information of C. sinensis in miRbase database, we first com-131 pare the clean data of the miRNA of the sample with the mature miRNA sequence of the 132 plant in miRbase v.21 database.” It’s not clear what species are they using to get the miRNAs of C.sinensis, maybe mature.fa file? In that case, this is not appropriate since it includes everything. More details should be provided in this step as it is essential for miRNA profiling.
- Line 145: Differential expression analysis, and KEGG and GO enrichment are not explained. In the case of DEA, which software or criteria have been used to get the p-value?
Minor concerns:
- Need further English editing
- Citation of all software tools that have been published should be addressed, some of them:
- Line 128 Rfam
- Line 129 Gen-Bank and BLAST
- Line 133: MiRBase
- Line 145: Mireap
- Line 149: psRNAtarget
- Line 156: ClusterProfiler
- Line 226: TPIA
- Line 179: Adapter
- Line 183: Read length
